# Inflammatory Changes after Medical Suppression of Suspected Endometriosis for Implantation Failure: Preliminary Results

**DOI:** 10.3390/ijms25136852

**Published:** 2024-06-22

**Authors:** Bruce A. Lessey, Allan Dong, Jeffrey L. Deaton, Daniel Angress, Ricardo F. Savaris, Stephen J. Walker

**Affiliations:** 1Department of OBGYN, Atrium Health Wake Forest Baptist, Winston-Salem, NC 27157, USA; adong@wakehealth.edu (A.D.); jldeaton@wakehealth.edu (J.L.D.); 2CiceroDx, Huntington Beach, CA 92649, USA; dan@cierodx.com; 3Department of OBGYN, School of Medicine, Universidade Federal do Rio Grande do Sul, Porto Alegre 90035-003, RS, Brazil; rsavaris@hcpa.edu.br; 4Wake Forest Institute for Regenerative Medicine, Winston-Salem, NC 27101, USA; swalker@wakehealth.edu

**Keywords:** endometriosis, implantation, infertility, recurrent pregnancy loss, inflammation

## Abstract

Unexplained euploid embryo transfer failure (UEETF) is a frustrating and unanswered conundrum accounting for 30 to 50% of failures in in vitro fertilization using preimplantation genetic testing for aneuploidy (PGT-A). Endometriosis is thought by many to account for most of such losses and menstrual suppression or surgery prior to the next transfer has been reported to be beneficial. In this study, we performed endometrial biopsy in a subset of women with UEETF, testing for the oncogene BCL6 and the histone deacetylase SIRT1. We compared 205 PGT-A cycles outcomes and provide those results following treatment with GnRH agonist versus controls (no treatment). Based on these and previous promising results, we next performed a pilot randomized controlled trial comparing the orally active GnRH antagonist, elagolix, to oral contraceptive pill (OCP) suppression for 2 months before the next euploid embryo transfer, and monitored inflammation and miRNA expression in blood, before and after treatment. These studies support a role for endometriosis in UEETF and suggest that medical suppression of suspected disease with GnRH antagonist prior to the next transfer could improve success rates and address underlying inflammatory and epigenetic changes associated with UEETF.

## 1. Introduction

Endometriosis is an inflammatory, estrogen-dependent disorder affecting up to 10% of reproductive-aged women [1,2]. Infertility and endometriosis are closely associated, noted in over 50% of women with this disease [3,4]. There remains a significant delay in diagnosis of endometriosis from 4 to 11 years [5,6] due in part to the dramatic decrease in laparoscopies now being performed for infertility indications [7]. When sought, endometriosis is discovered in women with unexplained infertility in a majority of cases [8,9,10,11,12], and treatment has been shown to improve fertility outcomes [9,13,14]. Despite this, laparoscopy is currently not recommended as part of a routine infertility evaluation [15].

Endometriosis and its accompanying inflammation affect endometrial function, leading to progesterone resistance [16,17,18]. The endometrium is a highly dynamic tissue that plays an essential role in establishment of pregnancy, with a window of implantation (WOI) occurring between 6 to 10 days after ovulation [19,20]. These changes, which are referred to collectively as “endometrial receptivity”, are complex and arise in response to ovarian steroids and related autocrine and paracrine factors [17,21]. In women with unexplained infertility or endometriosis, this window is often delayed or absent, leading to implantation failure or miscarriage [11]. Without clear diagnostic guidance, many women suffer with unexplained infertility or recurrent pregnancy losses without the benefit of proven or even recommended treatment options.

There is an increasing demand for Assisted Reproductive Technologies (ART), which offer the potential to improve fertility outcomes for many couples [22]. Despite the promise of these advanced technologies, many women undertaking IVF still encounter failure, with live births nationwide between 25 to 50%, in women under 40 (www.sartcorsonline.com, accessed on 1 June 2024). Even with the addition of Preimplantation Genetic Testing for Aneuploidy (PGT-A), success rates still range from 35–50%. Those who fail to conceive, despite transfer of a normal, euploid embryo, often have no identifiable reasons for their lack of success. As with unexplained infertility, these women with unexplained IVF failure often have undiagnosed endometriosis [23]. With new biomarkers, endometriosis has been shown to be common in women with unexplained infertility as well as those with IVF failure [24]. Identifying and providing preemptive treatment to those affected could improve outcomes through laparoscopic resection or medical suppression of disease [14].

Endometriosis likely contributes to implantation failure through epigenetically mediated changes in endometrial receptivity, associated with progesterone resistance [25]. A biomarker of endometriosis, BCL6, was shown to be associated with poor outcomes in in vitro fertilization (IVF) [24]. SIRT1, a histone deacetylase, has also been shown to be co-expressed with BCL6 and contribute to the inflammatory response associated with endometriosis [26]. As an estrogen responsive disease, ectopic endometriotic implants require estrogen; aberrant expression of the enzyme aromatase (CYP19a), maintains local estrogen production promoting disease proliferation. Aromatase expression and local estradiol production in both ectopic and eutopic endometrium are driven by inflammatory cytokines, including IL-6 and IL-17, among others [27,28]. This process stimulates cyclo-oxygenase activity, leading to increased levels of prostaglandin E2 (PGE2) and NF-kB signaling, which in turn leads to local estrogen action and progesterone resistance. This inflammation-driven cascade interferes with normal endometrial receptivity at the time of implantation (Figure 1a) [29,30].

Emerging strategies to improve outcomes for women with implantation failure involve suppression of suspected endometriosis prior to embryo transfer [14,31]. Protocols to suppress suspected endometriosis also vary widely, with differences in both treatment agents, timing and duration. In pursuit of better therapeutic options for women with unexplained implantation failure, we evaluated potential benefit of GnRH agonist suppression compared to “no treatment” in women with unexplained euploid embryo transfer failure (UEETF) and evaluated the presence of BCL6 and SIRT1, an associated histone deacetylase, in their endometrium prior to their next transfer. Based on these results, we performed a pilot randomized controlled trial using the orally active GnRH antagonist elagolix, compared to oral contraceptive pills (OCPs) for 2 months (EFFECT Trail; Elagolix for Fertility Enhancement Clinical Trial; NCT04039204) to compare inflammatory and epigenetic changes in blood in response to both treatments. These findings presented here support the hypothesis that negative pregnancy outcomes in IVF are associated with systemic inflammation and that suppression of suspected endometriosis, based on endometrial biopsy testing, may have a role in improving outcomes for this population of women.

## 2. Results

### 2.1. Rationale

Defects in endometrial receptivity due to inflammation and endometriosis have been proposed as a mechanism for both lack of implantation as well as pregnancy loss (Figure 1a). This is based, in part, on evidence that endometrial biopsy-based testing for BCL6 [32,33] and SIRT1 [26] is predictive for the presence of endometriosis in women with infertility.

#### 2.1.1. Non-Randomized Results

A survey of IVF cycles from 2019 to 2023 included 205 frozen embryo transfer (FET) cycles following PGT-A testing, including 127 first-time attempts (FirstPGTA) (Figure 1b). The demographics of each group are shown in Table 1. There were 61 cases with prior failure despite transfer of normal embryos with PGT-A testing. Of these, 48 (78.7%) consented for endometrial biopsy to rule out endometritis and test for BCL6 and/or SIRT1. HSCOREs for BCL6 testing were positive in 42 of 48 cases (87.5%), while HSCOREs for SIRT1 were positive in 39 of 45 cases tested (86.7%).

Of the sixty-one unexplained euploid embryo failure (UEETF) subjects who underwent subsequent monitored FET cycles with another single euploid embryo transfer, the 47 pretreated with GnRH agonist (GnRHa) for two months prior to transfer and had a significantly better ongoing pregnancy/live birth rate of 68.1% (32/47), compared to those with no additional treatment (NoTx; 35.7%; 5/14) (*p* = 0.02; *chi*-square) (Figure 1b).

#### 2.1.2. EFFECT Trial

EFFECT Trial (NCT04039204) enrolled subjects with failed euploid embryo transfers who had tested positive for both BCL6 and SIRT1 as described in the Section 4. The results are shown in Table 2. Two out of five in the OCP group had excessive bleeding and opted out of frozen embryo transfer. Based on intention to treat, the pregnancy rate was 60%. Two of three pregnant subjects experienced biochemical pregnancies resulting in an ongoing/delivered rate of only 20%. In the elagolix group, there was a 100% pregnancy rate (5/5), with two who suffered miscarriage and three with a successful delivery (live birth rate of 60%). Due to the small numbers of subjects enrolled, statistical significance was not obtained (*p* = 0.4, Fisher’s exact test).

#### 2.1.3. Inflammatory Array Results

EFFECT Trial participants had their blood sampled before and after treatment. Inflammatory array results for the OCP group are shown before and after treatment using PAX tubes (whole blood in RNA later). Gene (mRNA) expression profiling was performed on the nCounter Sprint^TM^ Profiler (Nanostring Technologies Inc., Seattle, WA, USA) using the nCounter Human Inflammation panel of 255 genes. A complete list of targets is included in the Appendix A. The overall results are provided in the Appendix A, but a heatmap of 10 differentially expressed transcripts (DETs) were generated based on *t*-test and significance level at *p* < 0.01 before and after suppression with OCPs (Figure 2a). Messenger RNA isolated from blood drawn in PAX tubes^®^ before and after elagolix treatment was likewise analyzed and those results presented in Figure 2b and Appendix A. In the elagolix group, 42 DETs were detected based on similar significance levels. Interestingly, the OCP group displayed both up and down-regulated inflammation targets, while the elagolix DETs were uniformly up-regulated after treatment and the two groups had few inflammatory DEGs in common. Principal component analysis (PCA) for OCPs vs. elagolix treatment is shown in Figure 2c,d.

In the OCP group, seven DETs were elevated before treatment including CCL11, IL12, CCL19 and GNGT1, while three DETs increased with treatment including TGFB1, IL10-RB, and MX2. In the elagolix group, 42 DETs were detected that all increased after suppression, compared to before treatment, including CSF2 and CSF3, CCL2, CCL8, CCL22, TWIST2, IL21, CCL19, FLT1, MMP3, CXCL10, PDGFA, IL13, IL11, CCL17, TGFB2 and TGFB3, CCL4 and ALOX15.

#### 2.1.4. MiRNA Array Results

MicroRNAs were also assessed on nCounter Sprint Profiler (Nanostring Technologies Inc., Seattle, WA, USA) using the nCounter Human miRNA Expression Assay kit (Human v3 assay) consisting of 827 miRNAs (see Appendix A). These data are presented in the Appendix A, and a heatmap of this comparison is shown in Figure 3a,b. The heat map for OCPs treatment displayed 10 differentially expressed transcripts (DETs) based on t-test and significance level at *p* ≤ 0.05 before and after suppression (Figure 3a), while the elagolix group had 337 differentially regulated miRNAs (Figure 3b). PCA plots for both OCPs and elagolix treatment are shown in Figure 3c,d, respectively.

## 3. Discussion

This study addresses, at least preliminarily, the question regarding how to address IVF failure following transfer of a euploid embryo. Recurrent implantation failure is considered by some a rare occurrence and report acceptable cumulative pregnancy after repeated transfers, without intervention [34]. Pirtea et al. [35] performed successive euploid embryo transfers after FET failure, and reported miscarriage rates were fairly constant in subsequent transfers, using euploid embryos, with a cumulative pregnancy rate of 92%. A criticism of this study was the high dropout rate and the supplementation of new subjects that dilute the results of subsequent transfers, making actual success rates difficult to ascertain.

Studies on Endometrial Receptivity Testing (ERA) have been popular and are based on the premise that the window of endometrial receptivity is short and variable, and that success rates could be improved by tailoring the timing of transfer [36]. Recent double-blind studies found that ERA testing with modification of transfer day did little to improve live birth rates [37]. Coutifaris demonstrated that histological dating was not related to fertility status [38], further bringing into question the ERA results. Finally, the study by Wilcox and colleagues had previously shown that the window of implantation is relatively wide [20].

The expanding use of PGT-A-defined “normal” embryos has focused attention back to the endometrium, which can represent a barrier to implantation, except during the WOI. High BCL6 expression in the endometrium is an indicator for the presence of inflammation and endometriosis, and predicts unexplained IVF failure [24,39]. We routinely employ GnRH agonist suppression before the next euploid embryo transfer to improve outcomes in women with UEETF. In the present study involving strict inclusion limited to unexplained euploid embryo failure subjects, we found statistically significant improvement using GnRHa alone for 2 months compared to a “no treatment” regime. While these findings are significant and may help expand treatment options for those who failed a subsequent FET, we were prompted to further investigate the basis for this improvement. We registered and performed a limited RCT comparing the better-tolerated, orally active GnRH antagonist, elagolix as a possible treatment option in women with UEETF, the EFFECT Trial. During that study, we noted a 100% implantation rate and a 60% life-birth rate in the elagolix group compared to a 20% live birth rate in the OCP control group. Importantly, this study afforded us the opportunity to obtain leukocyte-containing blood extracts (using PAX^®^ tubes) before and after both treatments to compare the inflammatory characteristics and miRNA changes after suppression in women with suspected endometriosis based on positivity of BCL6 and SIRT1 testing.

We employed the nCounter inflammation panel to examine changes in inflammation between the OCP and elagolix treatments. We report that in the OCP group, 7 DETs were elevated before treatment including IL12, CCL2, CCL11, CCL19, NOX1, MASP2, and GNGT1, while 3 DETs increased after treatment, including TGFB1, IL10-RB, and MX2. CCL11, also known as eotaxin-1, is an inflammatory biomarker that was reportedly elevated in peritoneal fluid and eutopic and ectopic endometrium of women with endometriosis [40], and induced in endometriotic stromal cells by interleukin-4 (IL-4) [41]. Importantly, it induces angiogenesis in endothelial cells, thus contributing to the pathogenesis of endometriosis [42]. We previously showed that CSF-2, aka GM-CSF, was elevated in plasma of women with endometriosis and decreased after surgery for this disease [43] and both CSF-2 and IL12 have been shown to be elevated in peritoneal fluid as well as endometrial and endometriotic cells of women with endometriosis [43,44,45]. CCL19 is also present in the peritoneal fluid of endometriosis cases and regulates proliferation and invasion of endometriotic cells via the PI3K/AKT pathway [46].

Following OCP treatment, we also noted increased expression of several interesting inflammation pathway members. Transforming growth factor-beta 1 (TGFB1) is an anti-inflammatory cytokine that has been shown to play a role in endometriosis pathogenesis [47,48] and suppresses or promote endometrial cancer depending on CD73 expression [49]. Its decreased expression after OCPs will require further investigation. Loss of CXCR4, the receptor for CXCL12, in response to OCPs is also interesting, as this has been reported to reduce proliferation and lesion number in endometriosis [50]. Blocking CXCL12/CXCR4 actions also decreases invasion and migration of the endometriotic 12Z cell line [51]. Finally, IL-23a and its association with the IL-17 axis promotes endometriosis and is involved in the pathogenesis of this disease [52]. It is reduced in response to 2 months of OCP use, suggesting a direct effect on this important and relevant inflammatory pathway.

In the elagolix group, 42 DETs were detected that all increased after suppression compared to before treatment, including IL22RA2, CSF2 and CSF3, CCL2, CCL8, CCL22, TWIST2, IL21, CCL19, FLT1, MMP3, CXCL10, PDGFA, IL13, IL11, CCL17, TGFB2 and TGFB3, CCL4 and ALOX15. Of these IL22 binding proteins, IL22RA2, increased after elagolix treatment is worthy of further investigation. IL22 in the setting of endometriosis promotes proliferation of ESCs via secretion of CCL2 and IL-8. CSF3 down-regulation by elagolix is also interesting, as it has been shown to be higher in PF of women with endometriosis [53]. Other moieties with anti-inflammatory roles in endometrial receptivity were elevated after elagolix treatment, including IL10, IL13, CCL18 and TWIST2 (see [54,55,56,57,58]). Genes associated with a receptive endometrium were also represented as elevated in response to 2 months of suppression with elagolix, including FLT1, IL11, CCL2 and CSF3 [59,60,61,62,63]. In addition, ALOX15, an essential enzyme for the metabolism of fatty acids, was elevated following elagolix treatment. ALOX15 has been reported to be induced by progesterone and likely essential for embryo implantation as shown in the mouse model [64], possibly indicating an improvement in progesterone resistance following medical suppression with elagolix.

Of note, there are several puzzles presented by these data. CCL19 is increased in blood from these volunteers, which has been found to be elevated in PF of infertile women with endometriosis [65]. Likewise, mRNA for IL17A was increased in blood following treatment for 2 months with elagolix.

MicroRNA expression levels were also assessed using nCounter Sprint Profiler (Nanostring Technologies Inc., Seattle, WA, USA) with the Human miRNA Expression Assay kit (Human v3 assay) consisting of 827 miRNAs. As shown in Figure 3a, a limited number of miRNAs were up- (*n* = 5) and down- (*n* = 5) regulated in the OCP treatment group. MiR-2110 was the most up-regulated miRNA and has been shown to be highly up-regulated in serum of healthy control women compared to PCOS [66]. PCOS is associated with subfertility, heavy and prolonged menses, progesterone resistance and endometriosis [67,68]. Mir-RNA-25-3p has been shown to be anti-inflammatory [69] and was up-regulated following two months of OCP treatment in our study. This miRNA is down-regulated in endometriosis [70] and targets SP1, a transcription factor involved in the pathogenesis of endometriosis and estrogen metabolism [71]. Finally, miRNA-204-5p was up-regulated in the OCP group; it promotes apoptosis by targeting BCL2 in prostate cells [72] and is a tumor suppressor in breast cancer [73], and lower levels predict lymph node negative status in endometrial cancers [74].

The miRNA array data for elagolix treatment was quite different from OCP results with most of the miRNAs being up-regulated by treatment with this GnRH antagonist (Figure 3b). The most up-regulated miRNA, miR-582-5p, targets CREB1/CREB5-NF-κB signaling [75] and is down-regulated in monocytes by opioid-induced immunosuppression. Its role in fertility and endometriosis or improved endometrial receptivity has not been examined. MiR-490-39 is 5-fold increased after elagolix treatment and has been shown to down-regulate NOTCH1 [76]. Interestingly, NOTCH1 is up-regulated by IL6 and inflammatory pathways in endometriotic lesions [77]. Elevated NOTCH1 is associated with progesterone resistance and lower progesterone receptor (PR) levels. Notch1 activity also mediates estrogen-stimulated stromal cell invasion in endometriosis [78].

Other pertinent up-regulated miRNAs of interest on this elagolix-treated array include miR-1205, acting like a tumor suppressor by inhibiting KRAS [79]. KRAS mutations play an important role in endometriosis invasion and pathogenesis [26,80]. Mir-520b targets PTEN in breast cancer cells and inhibits T cells and NK cells, and transforms macrophages toward the M2 type [81] and inhibits endothelial activation by targeting NF-kB transcription factor P65, and inflammation [82]. Finally, miR-509-3-5p was a previously shown miRNA up-regulated in endometriosis cases [83]. Of the few down-regulated miRNA in the elagolix treatment array, miR-142-3p was previously shown to modulate cell invasion and migration in colorectal cancers, and its down-regulation in suspected endometriosis cases by elagolix is open to speculation.

## 4. Materials and Methods

Electronic records of 205 frozen embryo transfer (FET) cycles using euploid embryos between 2019 and 2023 were examined. A total of 127 first IVF attempts (FirstPGTA) were evaluated and compared to 61 transfers in women with prior UEETF. We compared 47 of those who were treated in a non-randomized case series with long-acting GnRHa (Lupron^®^) suppression for 2 months (GnRHa) and 14 who received no treatment (NoTx) prior to the next euploid embryo FET. Pregnancy success was defined as live birth or ongoing pregnancy (heartbeat after 12 weeks).

To investigate these losses more fully, and using IRB-approved protocols, we procured endometrial biopsies from 48 women with prior UEETF, and performed standard immunohistochemistry (IHC) staining for BCL-6 and SIRT1 as previously described [24,26]. HSCORE was assigned as part of immunohistochemical (IHC) analysis, by a blinded observer as previously reported [83]. A cut-off of 1.4 and 2.0 (out of 4.0) was considered positive for BCL-6 and SIRT1, respectively. Pregnancy success rates are reported as a 95% CI and compared using *chi*-square for trend testing.

Ethics was approved for this study by the Institutional Review Board (IRB) for human subjects research at Atrium Health Wake Forest Baptist including endometrial biopsy and BCL6 and SIRT1 testing for failed euploid embryo transfer (Cellular Mechanisms of Infertility) and a randomized controlled study “Elagolix for Fertility Enhancement Clinical Trial” (Unexplained euploid embryo transfer failure). This RCT was registered on ClinicalTrials.gov (NCT04039204). Inclusion for this subset of women who failed a euploid embryo transfer using preimplantation genetic testing for aneuploidy (PGT-A), included age 18 to 42, parity G0 or greater, AMH >0.5 to <10, and each person was required to have failed a euploid embryo transfer with viable remaining euploid embryos for transfer. All subjects were required to be positive for endometrial BCL6 and SIRT1. Exclusion criteria included uterine fibroids >4 cm, polycystic ovary syndrome using Rotterdam criteria, ovarian failure, diabetes mellitus (Type I or II), untreated hypothyroidism, or elevated anticardiolipin and/or lupus anticoagulant abnormalities by history, hyperprolactinemia, uncorrected uterine anomaly, severe renal disease, osteoporosis, moderate or severe hepatic impairment defined by Child–Pugh classes B (moderate) or C (severe), women taking CYP3A inhibitors (e.g., ketoconazole) and women at high risk of thromboembolic disorders including smokers or those with cardiac valvular disease, atrial fibrillation or history of severe migraines. Endometritis on endometrial biopsy was exclusionary.

Prior to the next frozen embryo transfer (FET) using a remaining euploid embryo, subjects were randomized to receive either elagolix (200 mg twice a day (BID)) or oral contraceptives (OCPs) consisting of Orthocyclin for 2 months. Elagolix (Orilissa) is a new-generation FDA-approved orally active GnRH antagonist that is rapidly reversible, for the treatment of endometriosis and pelvic pain [84]. Following two months of treatment, subjects began a programmatic administration of estradiol followed by progesterone support before subsequent embryo transfer of a single euploid embryo. FET cycles were begun without intervening menstruation. Subjects began estrace 2 mg BID, plus estradiol patch every 3 days. Progesterone in oil IM injections were started when the lining was >6.5 mm and has a trilaminar appearance. Transvaginal ultrasound was performed between day 12 and 16. Transfer was performed 126 h after the start of progesterone, which was continued until 10 weeks of gestation if pregnant. If the human chorionic gonadotropin (hCG) test is negative, progesterone in oil administration was discontinued.

Human chorionic gonadotropin (hCG) levels were measured 7 to 8 days after the transfer twice over 2 days and followed with vaginal ultrasound at 7 weeks determination to document viability of a positive pregnancy test. Patients were followed for up to 9 months if pregnant and primary outcomes determined including: “cancellation”, “not pregnant”, “biochemical pregnancy or miscarriage” or “ongoing pregnancy/delivered” and “live birth rate”. Blood was obtained in redtop and purple top tubes for serum and plasma and PAX tubes (RNA later) at the beginning of the study and after the completion of the medication administration, and stored for future use.

Using intention to treat analysis, we randomized 10 UEETF subjects to this RCT. All patients had a prior endometrial biopsy and blood draw before initiating suppression therapy and at the conclusion of 2 months of treatment. Gene (mRNA) expression profiling was performed on the nCounter Sprint^TM^ Profiler (Nanostring Technologies Inc., Seattle, WA, USA) using the nCounter Human Inflammation panel of 255 genes. A list of differentially expressed transcripts (DEGs) was generated based on *t*-test and significance level at *p* < 0.01. Clinical outcomes were analyzed and compared using Fisher’s exact test. MicroRNAs were also assessed on nCounter Sprint Profiler (Nanostring Technologies Inc., Seattle, WA, USA) using the nCounter Human miRNA Expression Assay kit (Human v3 assay) consisting of 827 miRNAs.

### 4.1. RNA Isolation

Total RNA was isolated from whole blood, originally collected and stored in PAXGene tubes, using the QIACube Connect instrument (Qiagen, German Town, MD, USA) and reagents according to the manufacturer’s protocols. RNA quantity was assessed using a NanoDrop Spectrophotometer (ThermoFisher Scientific, Waltham, MA, USA).

### 4.2. Gene Expression Analysis

Total RNA (100 ng/per sample) was used in multiplexed assays to measure gene expression with either the nCounter^®^ Human Inflammation Panel (249 genes known to be differentially expressed in inflammation) or the nCounter^®^ Human miRNA Expression Kit (contains 827 human miRNA targets). Multiplex gene expression profiling was performed according to the manufacturer’s protocols using the nCounter^®^ Sprint Profiler instrument (NanoString Technologies Inc., Seattle, WA, USA), and the raw data files were uploaded to the web-based nSolver software (https://nanostring.com/products/ncounter-analysis-system/ncounter-analysis-solutions/, accessed on 10 June 2024) suite for evaluating QC metrics, data normalization, and differential gene expression analysis.

### 4.3. Variables

Sample was divided into 3 groups: FirstPGTA, GnRHa and No Treatment groups.

The following variables were analyzed: age (years), BMI (kg/m^2^), ongoing pregnancy (yes/no), race (white, black, Asian/other). Pregnancy was sub-divided into those with ongoing/delivered, and miscarriage/biochemical.

### 4.4. Statistical Analysis

Parametric data (age and BMI) were compared between 3 groups and the Kruskal–Wallis test was used for analysis. Fisher’s exact test was used to compare the pregnancy outcomes of individuals who used elagolix or OCPs. HSCORE was used as dichotomous data (using the cut-off 1.2 and 2.0 for BCL6 and SIRT1, respectively) and we calculated the percentage of positive cases in the whole population.

## 5. Conclusions

Based on non-randomized assessment of outcomes, we report that GnRH agonist suppression of endometriosis restored normal implantation and live birth rates compared to no treatment options for women with UEETF. In a pilot RCT comparing elagolix-mediated suppression (GnRH antagonist) for 2 months to OCP suppression, we documented differences in response. Elagolix dramatically changed the inflammatory milieu in blood compared to OCP suppression. A trend toward higher pregnancy rates and live birth rates was also reassuring and suggests that larger studies are warranted.

## Figures and Tables

**Figure 1 ijms-25-06852-f001:**
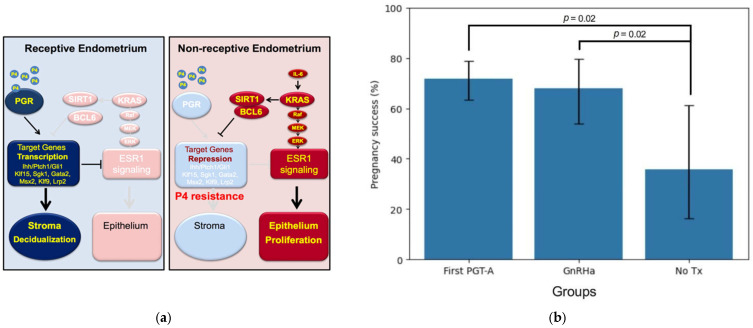
(**a**) Normal endometrial receptivity is largely regulated by progesterone which down-regulates estrogen receptors and initiates an extensive list of activated genes, along with stromal decidualization which is critical for successful pregnancy (left). In endometriosis, due to immune activation and inflammatory cytokines, progesterone action is suppressed and estrogen signaling amplified, contributing to infertility and pregnancy loss. (**b**) Pregnancy success rates (live birth and ongoing pregnancy) in first attempt PGT-A (First PGTA), compared to those with prior failure and GnRH agonist suppression for 2 months (GnRHa) or no further treatment (NoTx). The pregnancy success rate (95% CI) was 71.65% (63.27% to 78.77%), 68.09% (53.83% to 79.6%), and 35.71% (16.34% to 61.24%), respectively. No treatment was inferior to either GnRHa suppression or first time PGT-A success rates (*p* = 0.02, *chi*-squared).

**Figure 2 ijms-25-06852-f002:**
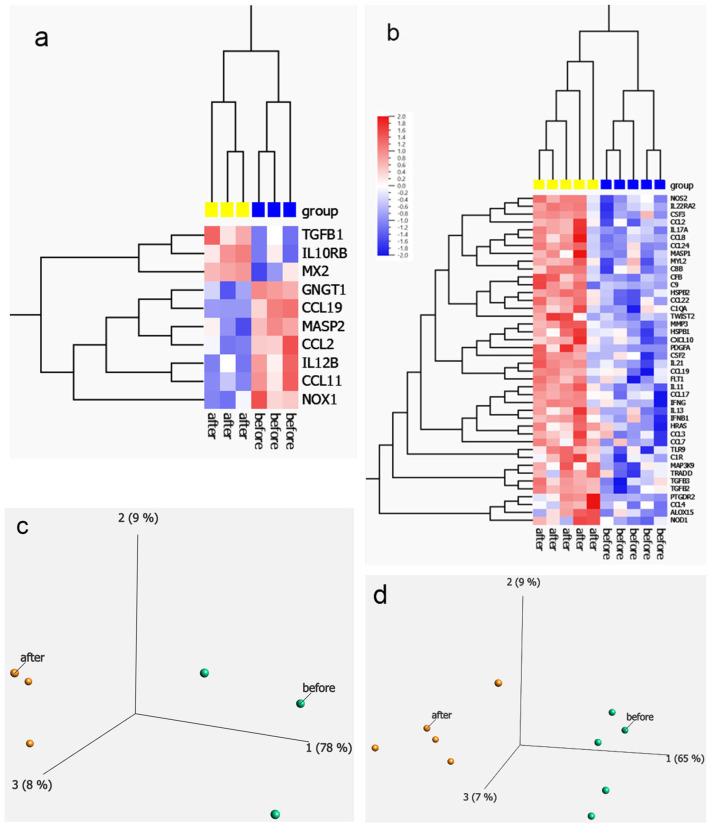
Differential expression of inflammation-related genes comparing pre- and post- treatment. (**a**) Heatmap of 10 differentially expressed transcripts (DETs; *p* ≤ 0.05) in peripheral blood detected using the Nanostring Inflammation panel before and after OCPs treatment. (**b**) Heatmap of 42 DETs (*p* ≤ 0.05) detected before and after elagolix treatment. Principal component analysis (PCA) plot for the OCP group (**c**) before and after treatment, and a PCA plot for the elagolix group (**d**).

**Figure 3 ijms-25-06852-f003:**
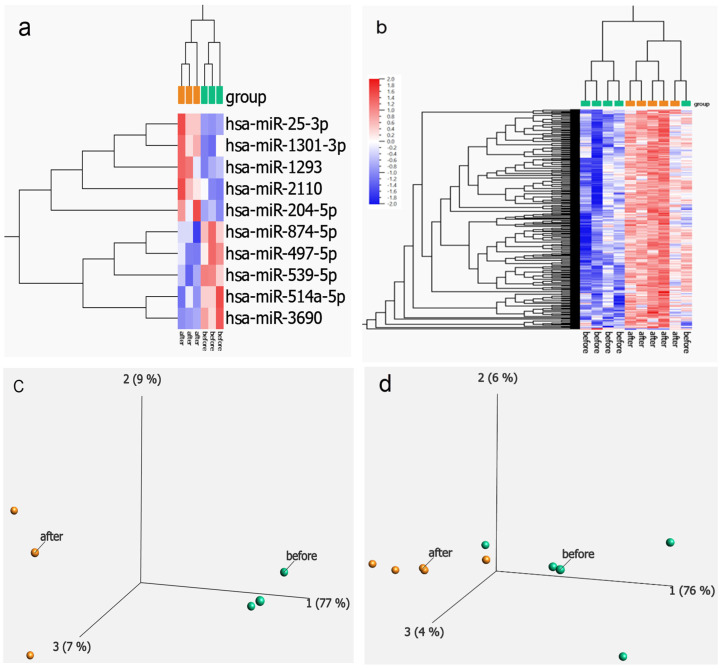
Differential expression of miRNAs comparing pre- and post- treatment. (**a**) Heatmap of 10 differentially expressed miRNA transcripts (DETs; *p* ≤ 0.05) in peripheral blood detected using the Nanostring Human miRNA panel before and after OCPs treatment. (**b**) Heatmap of 337 DETs (*p* ≤ 0.05) detected before and after elagolix treatment. Principal component analysis (PCA) plot for the OCP group (**c**) before and after treatment, and a PCA plot for the elagolix group (**d**).

**Table 1 ijms-25-06852-t001:** PGT-A Cycle Groups Demographics.

Demographics	FirstPGTA	GnRHa	NoTx	*p*
Age (median) (range)	36 (23–44)	35 (28–42)	37 (31–41)	0.3 ^a^
BMI (median) (range)	29 (17–43)	28 (20–40)	27.3 (18–40)	0.8 ^a^
Race	*n* = 127	*n* = 47	*n* = 14	0.07 ^b^
White	89	41	13	
Black	14	1	0	
Asian/other	24	5	1	

^a^ Kruskal–Wallis. ^b^ *chi*-square for trend.

**Table 2 ijms-25-06852-t002:** EFFECT Trial outcomes after 2 months treatment with elagolix or OCPs.

Outcome	Elagolix (*n* = 5)	OCP (*n* = 5)	*p* ^a^
Pregnant	5	3	0.4
Miscarriage/Biochemical	2	2	
Delivered/Ongoing	3	1	
Not pregnant	0	2	

^a^ Fisher’s exact test.

## Data Availability

The complete raw Nanostring data files will be shared on reasonable request to the corresponding author.

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
