# Peer review of "Inflammatory Changes after Medical Suppression of Suspected Endometriosis for Implantation Failure: Preliminary Results"

_ijms, 2024, doi:10.3390/ijms25136852_

Round 1
Reviewer 1 Report
Comments and Suggestions for Authors
In this manuscript, Lessey et al. performed a pilot randomized controlled trial comparing the orally active GnRH antagonist, elagolix, to oral contraceptive pill (OCP) suppression for 2 months before 2 months before next euploid embryo transfer, and examined the inflammation and miRNA expression in blood. The authors found that GnRH antagonist treatment prior to the next embryo transfer could improve pregnancy success rate.
Major concern:
1. Line 97, based on the data shown in Table 1, there are 126 first-time attempts (89 + 14 + 23= 126). Please explain this inconsistency.
2. Line 99, based on the calculation, the percentage of patients consented for endometrial biopsy should be 78.7% (48/61).
3. Line 102, based on the calculation, the percentage should be 86.7% (39/45).
4. Line 102, these presented data could not be found in Table 1.
5. Lines 106-107, the data shown here is not consistent with those shown in Figure 1b. For example, the rate in GnRHa group is 0.6809% as shown in the Figure.
6. For the heatmaps in Figures 2 and 3, a color scale bar should be added to show the fold change among different groups.
Author Response
- One race was listed as in the First PGTA group as unknown and not counted in our pivot table; this was corrected in Table 1 (89+14+24 = 127).
- The reviewer is correct; 78.7%, not 78.6%. This was a rounding error and has been corrected.
- Likewise this percentage was incorrect and we appreciate the opportunity to correct that on line 102.
- We noticed that during this review as well and removed the text suggesting ('see table 1). Thank you.
- We are not sure what the reviewer is referring to here; there were 61 UEETF subjects; 47 received GnRHa and 14 received no treatment. The success rates (ongoing pregnancy rate/live birth rate was 32/47 or 68.1% for GnRHa and 5/14 (35.7%) for the no treatment. This is what is presented in the text and Figure 1b.
- We added the request key for color in the heat map figures.
Reviewer 2 Report
Comments and Suggestions for Authors
The study investigates the comparison of elagolix and oral contraceptive pills (OCPs) to increase the success of pregnancy outcomes in IVF for women suffering from endometriosis. This research is both interesting and significant in the field of infertility.
However, I do some concerns:
1. Due to the small number of subjects and the lack of statistically significance differences between the studied groups, the authors should clearly state in the title, that the study should be treated as preliminary.
2. In verse 86, the underline should be removed.
3. In my opinion the results section is mixed with the Material&methods section (e.g. the sentence in verse 104-105: „Forty-seven were pretreated with GnRH agonist (GnRHa) for two months prior” should be introduce in the Materials&methods section, not in the results section). Therfore, these sections should be rewritten.
4. Did the authors assess inflamatory parameters other than SIRT1 mentioned in the text, such as: IL12, CCL2, CCL11, CCL19, NOX1, MASP2, GNGT1, TGFB1, IL10-RB, or MX2 in the current studied groups? Please provide more details on why SIRT1 was chosen to analyze inflammation.
5. Verse 185-186: „Some clinics report cumulative pregnancy after repeated transfers, 185 without intervention” – a reference is needed.
6. Verse 194: „Surprisingly few studies have directly addressed potential treatments for UEETF” – reference is needed
7. In my opinion, the Discussion section should be more focused on the obtained results.
Author Response
- The title was changed to Inflammatory Changes after Medical suppression of suspected Endometriosis for Implantation Failure: Preliminary Results
- Underlining was removed
- The methods portions of the results section was rewritten to exclude duplication of later Methods (see highlighted version).
- This study was funded by the NIH R44 mechanism specifically to look at women who co-expressed BCL6 and SIRT1, which are biomarkers of endometriosis. This association of Bcl6 and Sirt1 was not clearly stated so we added this sentence to the Introduction and changed the order of the references, where appropriate.: "SIRT1, a histone deacetylase, has also been shown to be co-expressed with BCL6 and contribute to the inflammatory response associated with endometriosis [26]".
- We appreciate that comment and have added an reference by Richard Scott.
- That sentence was removed
- We rewrote the discussion and eliminated excess (non-relevant) paragraphs and reordered the references.
Round 2
Reviewer 1 Report
Comments and Suggestions for Authors
1. In Table 1, one number is still not correct. The adjusted number should be 24.
2. In Figure 1b, the authors’ calculation is accurate. However, the presentation was not correct. Based on the Figure 1b, the pregnancy success rate was 0.7165%, 0.6809%, and 0.3571%, respectively.
3. In Figure 2b, based on the color scale bar that the authors added, it seems like that the data in the heat map shown in Figure 2b is not consistent with those presented in Table S3. For instance, in the Figure 2b, the fold change of NOS2 expression should be below 2. However, its fold change in Table S3 is more than 2. Additionally, the same issue exists for other heat maps in Figure 3. Please clarify this inconsistency.
Author Response
- We stand corrected; thank you.
- We changed Figure 1b to reflect pregnancy rate (%) from 0 to 100%. Thank you for catching that. Also added to the figure legend the following: "The pregnancy success rate was 71.65%, 68.09%, and 35.71%, respectively"
-
The heatmap in Figure 2b displays a representation of the expression level of each individual gene in each of the 10 samples. The color scale bar is meant to give context to the relative expression level between individual samples, rather than to compare fold change between the 2 groups. Therefore, while it may appear that, for example, none of the individual NOS2 values shows an expression level ≥2, the average expression levels of the “after treatment” samples (N=5) is 2.1-fold higher than the average expression “before treatment” (N=5).